# Single-cell microRNA sequencing method comparison and application to cell lines and circulating lung tumor cells

Sarah M. Hücker [1,8], Tobias Fehlmann [2,8], Christian Werno[1], Kathrin Weidele[1], Florian Lüke[1,3], Anke Schlenska-Lange[4], Christoph A. Klein [1,5], Andreas Keller [2,6,7,9] & Stefan Kirsch [1,9 ✉]

Molecular single cell analyses provide insights into physiological and pathological processes. Here, in a stepwise approach, we first evaluate 19 protocols for single cell small RNA sequencing on MCF7 cells spiked with 1 pg of 1,006 miRNAs. Second, we analyze MCF7 single cell equivalents of the eight best protocols. Third, we sequence single cells from eight different cell lines and 67 circulating tumor cells (CTCs) from seven SCLC patients. Altogether, we analyze 244 different samples. We observe high reproducibility within protocols and reads covered a broad spectrum of RNAs. For the 67 CTCs, we detect a median of 68 miRNAs, with 10 miRNAs being expressed in 90% of tested cells. Enrichment analysis suggested the lung as the most likely organ of origin and enrichment of cancer-related categories. Even the identification of non-annotated candidate miRNAs was feasible, underlining the potential of single cell small RNA sequencing.

[1] Division of Personalized Tumor Therapy, Fraunhofer-Institute for Toxicology and Experimental Medicine, Regensburg, Germany. [2] Chair for Clinical Bioinformatics, Saarland University, Saarbrücken, Germany. [3] Department of Internal Medicine III Hematology & Oncology, University Medical Center Regensburg, Regensburg, Germany. [4] Department of Oncology and Hematology, Hospital Barmherzige Brüder, Regensburg, Germany. [5] Experimental Medicine and Therapy Research, University of Regensburg, Regensburg, Germany. [6] Center for Bioinformatics, Saarland Informatics Campus, Saarbrücken, Germany. [7] Department of Neurology and Neurological Sciences, Stanford University, Stanford, CA, USA. [8] These authors contributed equally: Sarah M. Hücker, Tobias Fehlmann. [9] These authors jointly supervised this work: Andreas Keller, Stefan Kirsch. ✉email: stefan.kirsch@item.fraunhofer.de

Tens of thousands of small non-coding RNAs (sncRNAs) such as piRNAs, miRNAs, small nucleolar RNAs (snoR-NAs) are transcribed from the genome. Although they are not translated to proteins, these molecules are integral contributors to physiological and pathological processes. Among the best-studied sncRNAs are microRNAs (miRs, miRNAs). MiRNAs are important posttranscriptional regulators that are evolutionary very well conserved: The 20–25-nt long molecules bind to complementary regions on target mRNAs, which leads to mRNA degradation or translational repression[1]. Over 60% of human genes contain miRNA binding sites[2]. The most recent release of the miRBase (v22) lists 2654 annotated human miRNAs[3]. In addition to the miRBase, several other miRNA databases list, however, more specific or sensitive miRNA sets[4], and the total number of human miRNAs is estimated to be in the range of 2300 miRNA[5]. In pathologic conditions such as cancer, the transcription of many miRNAs is altered, which in turn changes the abundance of target mRNAs. Therefore, miRNAs have great potential as diagnostic or prognostic biomarkers. Even more, they represent promising novel drugs or therapeutic targets[6]. However, these promising clinical applications of miRNAs require methods for the accurate and reproducible quantification of global miRNA expression.

For bulk sncRNAs sequencing, a wide range of protocols and commercial kits exists. The principle of these methods is based on either sequential adapter ligation or polyadenylation of the miRNAs. Several studies comprehensively compared the performance of these methods[7–13]. However, already at high input concentrations biases were evident: The adapter ligation efficiency varies 1000-fold depending on miRNA sequence and secondary structure, which leads to low quantification accuracy[14]. Improved ligation conditions and adapters with random nucleotides can reduce this bias[8,9,15]. In addition, polyadenylation-based protocols are influenced by miRNA sequence and other RNA species could also be polyadenylated[8]. Moreover, the miRNA libraries contain adapter dimers that are difficult to separate from informative reads due to the small size of miRNAs. The adapter dimer problem increases with low input samples, because a high excess of adapters is required for efficient ligation[16]. Chemically modified adapters or removal of excess adapters reduced the amount of adapter dimers[17].

It is reasonable to hypothesize that respective experimental biases substantially increase if the input amount is decreased to single cell level. As single cell mRNA studies are widely applied to uncover insights into processes such as cellular differentiation or adaptation, single cell miRNA studies lag behind. Of note, not even the total amount and number of different miRNAs expressed in a single cell is exactly known. However, similar to scRNA-Seq, single cell miRNA-Seq would add to our understanding of molecular regulatory processes. Moreover, in order to study rare cell populations such as circulating tumor cells (CTCs), a single cell miRNA sequencing protocol is mandatory. The Sandberg group published two single cell miRNA-Seq protocol versions[18,19], but other miRNA-Seq protocols for low input samples are also available[9,20,21]. However, no comprehensive comparison of the approaches on single cell level is available yet.

Therefore, in this study, 19 miRNA-Seq protocol variants using defined samples with very low input were evaluated regarding their accuracy, reproducibility, and major sources of bias. The best performing protocols were then used at the single cell level and the same quality parameters were determined. Finally, we show the applicability of a selected protocol to a broad range of different cell lines and even clinical samples by analyzing the miRNA profiles of single CTCs of seven small cell lung cancer (SCLC) patients.

## Results

**Experimental design**. Towards a stable single cell sequencing of miRNAs, we implemented a three-stage approach (Fig. 1a). In the first stage, we comprehensively tested four ligation-based protocols later referred to as SB[18], SBN[19], CL[20], and 4N[9]. In addition, one polyadenylation-based miRNA-Seq protocol (CATS[21]) was included. Remarkably, we tested 19 variations of the four protocols with adapted experimental parameters: The adapters were exchanged between protocols, the adapter ligation time was increased (16 C[8]), a 5′ adapter with a complementary sequence to the 3′ adapter was designed (C3[15]), the unique molecular identifier (UMI) was shortened to 6 nt (UMI6), and an oligonucleotide to block reverse transcription of the adapter dimer was tested (Block). Details on the 19 evaluated protocol variants are provided in Supplementary Fig. 1 and Supplementary Table 1. In this first stage, we sequenced triplicates of single cells of the breast cancer cell line MCF7 spiked with 1 pg of miRXplore Universal Reference. This standard consists of 1006 miRNAs and artificial sequences from different species in equimolar concentration (Supplementary Data 1). We selected the eight best-performing protocols regarding their amount of adapter dimers, reads mapping to miRNA, number of detected miRXplore miRNAs, reproducibility, and quantification accuracy. We confirmed our findings by sequencing three additional replicates.

In the second stage, we evaluated these eight protocols on the single cell level. This time, we sequenced six replicates of MCF7 single cell equivalents to reduce the cell-to-cell variability. From these experiments, we selected the best protocol that showed a low amount of adapter dimers, a high amount of reads mapping to miRNAs, a high number of detected miRNAs, and high reproducibility between replicates.

In the third stage, we applied the best protocol to sequence six single cells each of eight different cell lines (fibroblasts, T-cells, monocytes, macrophages, lymphoblasts, colorectal adenocarcinoma, lung cancer, and hepatocellular carcinoma) and 67 single EpCAM positive CTCs from the blood of seven SCLC patients.

**Performance of miRNA-Seq strongly depends on the applied protocol**. Initially, we tested 19 miRNA-Seq protocol variants in triplicate using single MCF7 cells spiked with miRXplore. As a first quality control parameter, we measured the DNA concentration of the final libraries. The concentrations were variable, ranging from 0.39 ng μl$^{-1}$ ± 0.03 ng μl$^{-1}$ (protocol CL_UMI6) to 42.2 ng μl$^{-1}$ ± 0.65 ng μl$^{-1}$ (protocol SBN_4N; Supplementary Table 2). The fragment length distributions also showed differences. Products of around 125 bp should represent adapter dimers, products of around 145 bp represent libraries with miRNA inserts, and products larger than 155 bp are likely to represent inserts of other longer RNA types, e.g., lncRNA, mRNA, or snoRNA (Supplementary Fig. 2). Even though we quantified the libraries by qPCR prior to sequencing and pooled them in equimolar amounts, the number of sequenced reads varied from 200,000–2,650,000 (Supplementary Fig. 3). We computationally removed reads shorter than 18 nt and low-quality reads. Again, the results showed strong variations: Out of 19 tested protocols, six yielded over 90% of reads to be excluded, which disqualified the respective protocols from further analyses (Fig. 1b). In contrast, the best performing protocols had mapping rates of 60% to the human genome. Notably, about 10% of the reads mapped at multiple loci (Supplementary Data 2). Finally, between 10–50% of total reads matched annotated miRNA loci. The other mapped reads mainly match to protein-coding genes, intergenic regions, long non-coding RNAs (lncRNAs), or snoR-NAs; other RNA types can be neglected. The protocols 4N, 4N_C3, 4N_CL, SB, SB_4N, CL, SBN, and SBN_CL detected

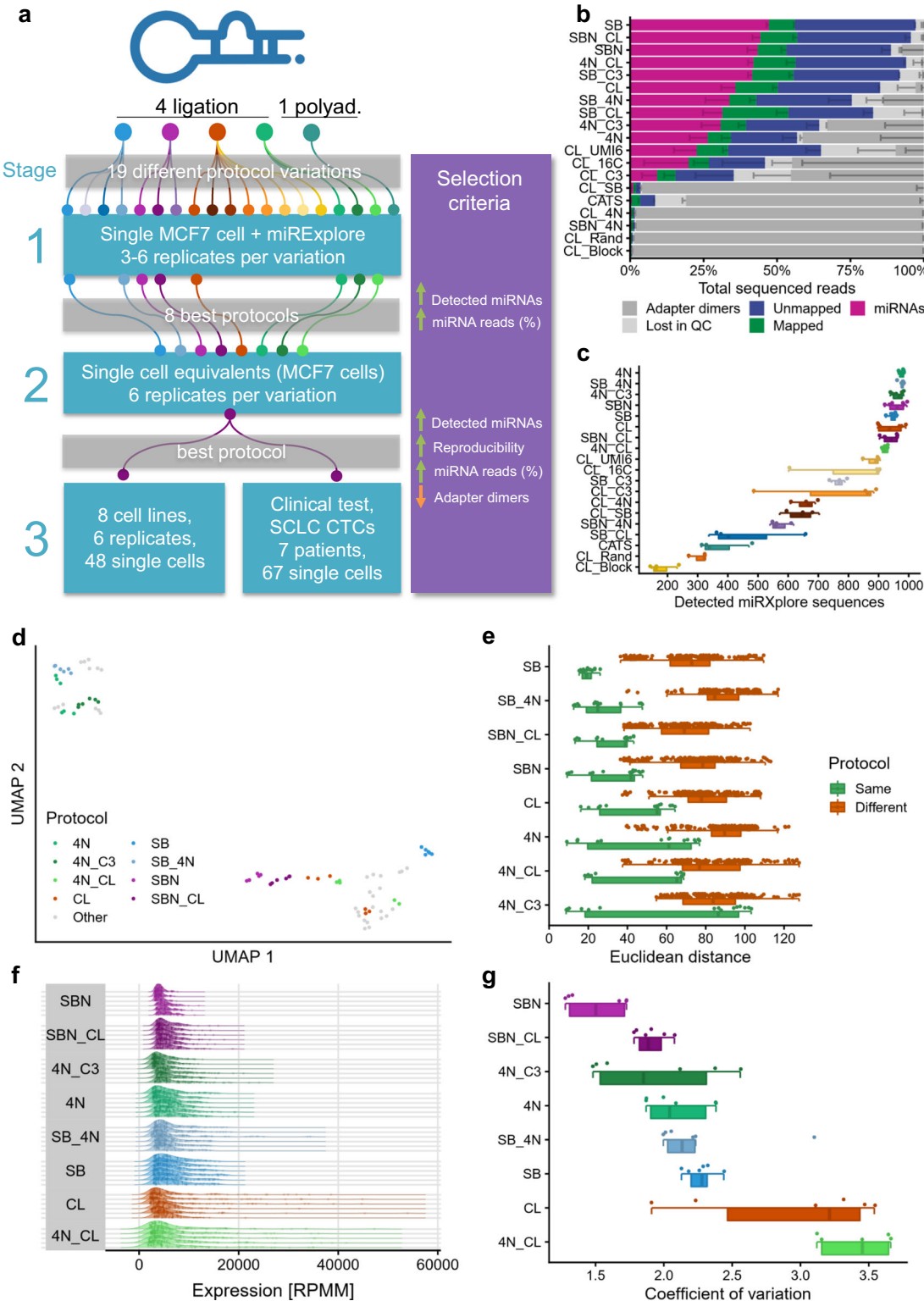

almost all 1006 miRXplore spike-in sequences (Fig. 1c). Interestingly, nine of ten sequences detected in less than 10% of all samples across all protocols are artificial calibration nucleotides, and only hsa-miR-193a-3p was detected in a similarly few samples (6.3%; Supplementary Data 3). The number of additionally detected human miRNAs, that are not part of the spike-in, is low (1–22 miRNAs), indicating that single MCF7 cells contain a much lower miRNA concentration than the 1 pg spike-in. If the reads are mapped to the miRXplore spike-in, good performing

protocols had over 90% of their reads mapped to the standard (Supplementary Fig. 4).

We selected the eight protocols with the highest number of reads mapping to miRNA loci and highest number of detected spike-in sequences (on average at least over 900), processed three additional replicates, and analyzed these protocols in further detail. First, we performed a dimension reduction analysis via UMAP which showed that samples cluster according to the used protocol. We observed that samples processed with the 5′ and 3′

**Fig. 1 Experimental setup and protocol comparison with miRXplore spike-in (stage 1). a** Overview of the experimental setup consisting of three stages. SCLC CTCs = small cell lung cancer circulating tumor cells. **b** Read distribution for all tested protocols, sorted by miRNA reads proportion. The data are presented as mean values ± the standard deviation ($n = 6$ biologically independent samples for the top eight protocols, $n = 3$ for the others), which is shown as a smaller error bar in a darker color than its corresponding read group and only represented in one direction. **c** Detected miRXplore sequences for all tested protocols, sorted by decreasing average per protocol shown as boxplot (bottom) and dot plot (top). Each sample is shown as one dot and colored by protocol. The boxes span the first to the third quartile with the vertical line inside the box representing the median value. The whiskers show the minimum and maximum values or values up to 1.5 times the interquartile range below or above the first or third quartile if outliers are present. **d** UMAP embedding of all sequenced samples with miRXplore spike-in. The samples of the best eight protocols are highlighted in their respective color. The remaining protocols are grayed out. **e** Euclidean distance on the $\log_2$ transformed sequence expression showing the reproducibility between all replicates of the same protocol (green) and between all samples of one protocol variant compared to all other protocol variants (brown). Each dot represents the distance observed between two samples. Only nonredundant distances are shown (i.e., the distance of sample 1 to sample 2 is considered identical to the distance of sample 2 to sample 1). For each protocol, a dot plot (top), as well as a boxplot (bottom), is shown. The boxplot was defined in the same manner as for panel **c**. **f** Distribution of the top 100 highest expressed miRXplore sequences per sample, normalized as reads per million mapped (RPMM). The samples are grouped by protocol and ordered by ascending coefficient of variation. The vertical lines inside the areas delimit the quartiles. Every dot inside the area represents the expression level of one sequence. **g** Coefficient of variation for all samples grouped by the protocol in ascending order shown as dot plot (top) as well as boxplot (bottom). Each sample is represented by a dot. The boxplot was defined in the same manner as for panel **c**. Source data are provided in the Source Data file.

4N adapters showed a clear split in comparison to the other protocols (Fig. 1d). In the next step, we evaluated the reproducibility of the measurements of each sample and found that the SB protocol showed the highest reproducibility (lowest Euclidean distance between the replicates of the same protocol), followed by the SB_4N and SBN_CL protocols, while the 4N protocols (4N, 4N_CL and 4N_C3) showed the lowest reproducibility (Fig. 1e). A comparison of the replicates of one protocol to all other protocols highlighted that the samples of protocol 4N had the highest Euclidean distance, i.e., were the most different from all other protocols. It is important to interpret the results in the light of spiked-in miRNAs, which should have the same concentration within and across protocol variants. An analysis of the nucleotide content of the miRXplore sequences, as well as the minimum free energy of their secondary structures, showed that only the G-content seemed to influence the detection rate in all protocols (Spearman correlation of 0.45, $P = 2.8 \times 10^{-52}$), with an increasing G-content leading to an increasing detection probability (Supplementary Fig. 5 and Supplementary Data 3). For all protocols, we observed large variations in the miRXplore miRNA expression levels. Already the top 100 most expressed spike-ins show differences of several orders of magnitude (Fig. 1f). Protocol 4N_CL showed the largest variance from the expected expression values with a coefficient of variation of 3.407 and protocol SBN showed the lowest variance, and therefore the best accuracy with a coefficient of variation of 1.506 (Fig. 1g). Since six of the eight best protocols also provide UMI sequences, we deduplicated the read mappings and evaluated their variation on the remaining reads (Supplementary Fig. 6). The protocol 4N_CL remained the one with the largest variance, with a coefficient of variation of 2.170, while the protocol 4N_C3 showed the least variation (coefficient of variation of 1.046). However, some human spike-in miRNAs could be expressed by the MCF7 cells as well and might increase the observed variance as a background signal. To avoid biases due to different sequencing depths, we performed our analyses on subsampled samples with 300,000 reads as well, which confirmed the observed patterns (Supplementary Fig. 7).

Overall, we applied 19 different protocols for miRNA analysis of very low input samples. Our comprehensive evaluation based on the miRXplore Universal Reference indicated that eight protocols performed best. Only ligation-based approaches were among the list of top-performing protocols such that the polyadenylation-based approach (CATS) was not contained in the second stage.

**The SBN_CL protocol shows high reproducibility and detects most miRNAs.** In the second stage, the eight best protocols were analyzed using single cell equivalents of the MCF7 cell line. Similar to the first stage, the DNA concentrations of the libraries showed high variability and ranged from 1.37 ng µl$^{-1}$ ± 0.13 ng µl$^{-1}$ (CL) to 36.53 ng µl$^{-1}$ ± 23.61 ng µl$^{-1}$ (4N_C3; Supplementary Table 3). The fragment length distribution was also comparable to the spike-in experiment with an increased number of small fragments (Supplementary Fig. 8). Likewise, the number of sequenced reads was comparable with the first stage and varied between 195,000–1,400,000 reads (Supplementary Fig. 9). The total number of reads remained comparable from the first to the second stage (Fig. 2a). However, the fraction of reads mapping to the human genome and the frequency of reads mapping to miRNAs clearly decreased (Fig. 2b). In case of the protocols SBN, SB_4N, 4N, and 4N_C3, less than 10% of the total reads mapped to the human genome, and the libraries consisted almost completely of adapter dimers. Compared to the spike-in experiments in stage 1, the amount of adapter dimers increased for all protocols 2- to 399-fold (Fig. 2c). Furthermore, the amount of multi-mapped reads increased to 34–73% of mapped reads (Supplementary Data 4). Finally, a maximum of 2.0% of total reads and 3.0% of mapped reads (SBN_CL) matched to annotated human miRNAs. Still, miRNAs were detected in all protocols: We monitored on average 55 to 178 different miRNAs between the protocols (Fig. 2d). One of the replicated SB libraries even contained 327 different miRNAs. The other replicates, however, yielded only between 134 and 189 miRNAs. The SBN_CL protocol showed the highest concordance with on average 178 miRNAs (SD 21.3) per replicate. Next, we performed a dimension reduction analysis via UMAP (Fig. 2e), which showed that most replicates clustered according to their protocol as the major driving factor, followed by the sequencing run (the six replicates were sequenced in two batches of three replicates). As for the UMAP dimension reduction for the protocols of the first stage, we observed again a split between protocols with 5′ and 3′ 4N adapters in comparison to all others, although this split was less pronounced. An evaluation of the reproducibility of the measurements of each sample highlighted that the replicates of the SBN_CL protocol had the highest reproducibility (lowest Euclidean distance between replicates of the same protocol), followed by the SB protocol (Fig. 2f). The CL protocol was found to be the one with the lowest reproducibility, i.e., the highest Euclidean distance between the replicates. An evaluation between single protocols in comparison to all others showed that all protocols

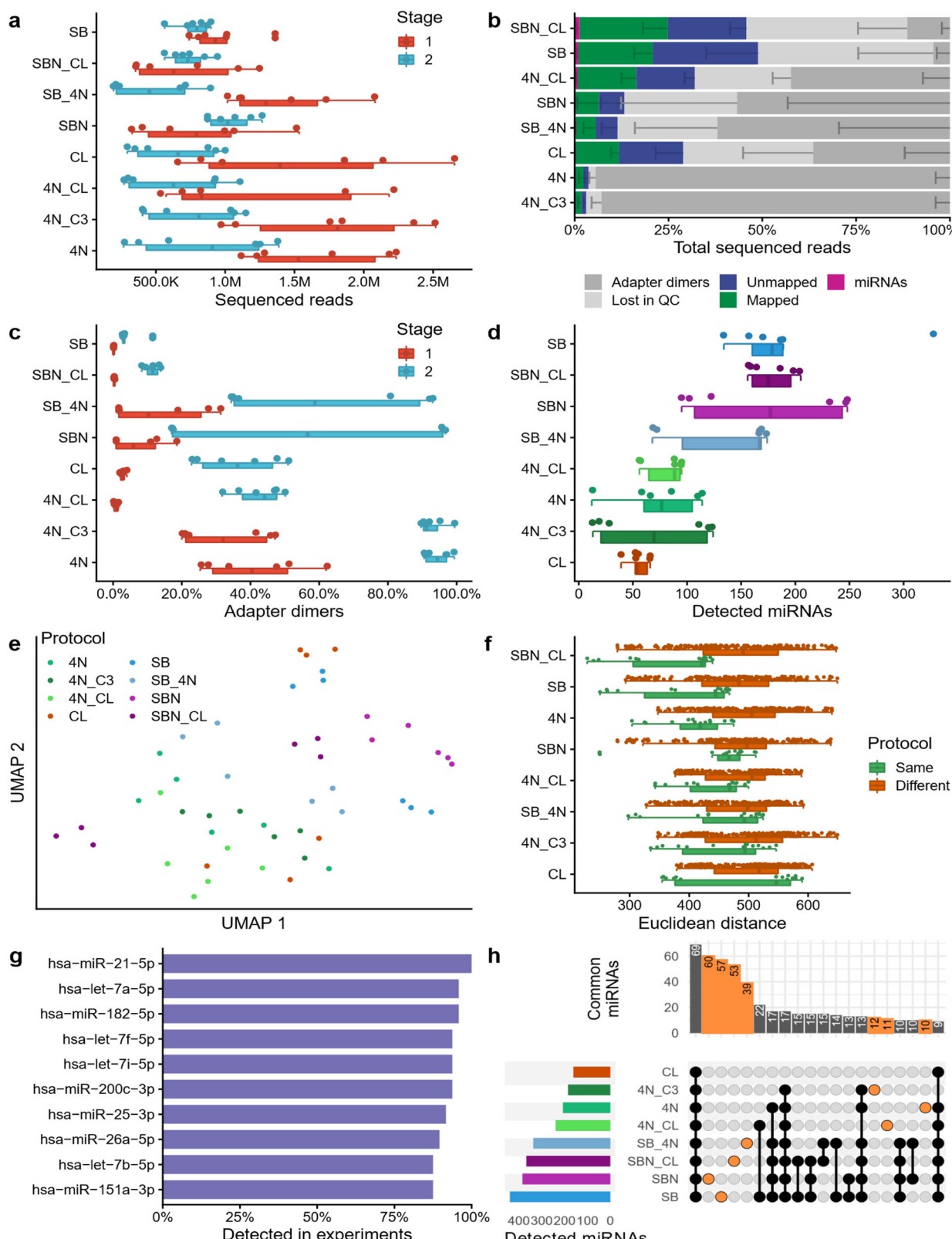

were similarly different from each other, while the samples of protocol CL showed on a median the largest difference. Focusing on the detected miRNAs, we observed the same trends. In total, up to 713 miRNAs were present in at least one replicate of the tested protocols. MiR-21-5p was found in all sequenced libraries followed by let-7a-5p and miR-182-5p (Fig. 2g). Finally, a set analysis comparing the overlaps of all detected miRNAs per

protocol was performed. This underlined the high heterogeneity of miRNAs detected per experimental setup. While 69 miRNAs were detected in at least one replicate in all protocols, 60 miRNAs were exclusively detected by the SBN protocol, followed by the SB protocol with 57 exclusive miRNAs and the SBN_CL protocol with 53 exclusive miRNAs (Fig. 2h). Analogous to the first stage, we performed our analyses on subsampled libraries with

**Fig. 2 Protocol comparison with single cell equivalents (stage 2). a** Number of reads sequenced in stage 1 (miRXplore spike-in) and 2 (MCF7 single cell equivalents), shown as boxplot (bottom) and dot plot (top). Each dot represents one sample. The boxes span the first to the third quartile with the vertical line inside the box representing the median value. The whiskers show the minimum and maximum values or values up to 1.5 times the interquartile range below or above the first or third quartile if outliers are present. **b** Read distribution for all tested protocols, sorted by miRNA reads proportion. The data were presented as mean values ± the standard deviation ($n = 6$ biologically independent samples), which is shown as a smaller error bar in a darker color than its corresponding read group and only represented in one direction. **c** Adapter dimers found in stage 1 and stage 2, shown as boxplot (bottom) and dot plot (top). Each dot represents one sample. The boxplot was defined in the same manner as for panel **a**. **d** Detected miRNAs for every sample, sorted by decreasing average per protocol shown as boxplot (bottom) and dot plot (top). Each sample is shown as one dot and colored by protocol. The boxplot was defined in the same manner as for panel **a**. **e** UMAP embedding of all samples. Each sample (dot) is colored by its protocol. **f** Euclidean distance on the $\log_2$ transformed sequence expression showing the reproducibility between all replicates of the same protocol (green) and between all samples of different protocols (brown). Each dot represents the distance observed between two samples. Only nonredundant distances are shown (i.e., the distance of sample 1 to sample 2 is considered identical to the distance of sample 2 to sample 1). For each protocol a dot plot (top), as well as a boxplot (bottom), is shown. The boxplot was defined in the same manner as for panel **a**. **g** Top ten miRNAs detected in multiple experiments. **h** Upset plot showing the miRNAs jointly detected by multiple protocols, or exclusively found in only one protocol (orange). The bar plot at the top shows on the y-axis the number of miRNAs detected by the protocols highlighted by connected black or orange dots in the grid below. The bar plot on the left shows on the x-axis the total number of miRNAs detected in the least one of the replicates of the protocol shown on the y-axis. Source data are provided in the Source Data file.

300,000 reads and confirmed the same patterns (Supplementary Fig. 10).

In conclusion, the protocols SB and SBN_CL showed the best results in our experiments on the single cell level. Due to its higher reproducibility, the SBN_CL protocol was selected for the miRNA analysis of eight cell lines and patient-derived CTCs in the third stage.

**SBN_CL protocol shows comparable performance in eight different cell lines**. In order to investigate if the SBN_CL protocol shows robust performance in different cell types, we analyzed six single cells each of the following cell lines: epithelial cancer cells (liver HepG2, lung A549, colon HT29), hematopoietic cancer cells (monocyte THP-1, T-cell Jurkat, macrophage KG1, lymphoblast REH), and healthy BJ fibroblasts. The SBN_CL protocol worked in all different cell types. The overall protocol performance is comparable to the MCF7 single cell equivalents. On average 30.5% (SD 8.5%) of total reads could be mapped to the human genome and 1.3% (SD 0.76%) matched annotated miRNAs (Fig. 3a and Supplementary Data 5). Per single cell, 32–255 different miRNAs could be detected (median 174; Fig. 3b). Dimension reduction via UMAP showed that the samples cluster by cell type, whereas the hematopoietic cell lines cluster closer together (Fig. 3c). Detailed analysis of the detected miRNAs shows that 128 miRNAs could be detected in at least one replicate of all eight cell lines and 16 (Jurkat) − 42 (HepG2) miRNAs were specific to a certain cell line (Fig. 3d). To determine if the miRNA profiles observed in the cell lines showed patterns in line with the literature, we performed a pathway enrichment analysis with miEAA 2.0[22] for the miRNAs of each cell line, sorted by decreasing mean expression. All profiles yielded enrichments typical for the studied cell line, i.e., for A549 and BJ we found significant enrichment for lung and skin tissue, respectively (Fig. 3e, f), for HepG2, HT29, KG1, REH, and THP-1 we found significant enrichment for their specific diseases (Fig. 3g–l). In summary, the sc-miRNA-Seq SBN_CL protocol can be used to determine tissue- and/or disease-specific miRNA profiles in a variety of different cell types.

**miRNA profiles of SCLC patient CTCs show high intrapatient variability**. In seven SCLC patients, single CTCs were isolated from blood using EpCAM staining (Fig. 4a). For these patients, in total 67 CTCs were sequenced using the SBN_CL protocol. Additionally, two negative controls containing only reagents without a cell were tested. The number of EpCAM positive cells per patient varied between 2 and 28 cells (Fig. 4b). For the patient

CTCs, we sequenced between 370,000 and 700,000 reads per cell, of which on average 62.7% (SD 6.6%) were lost in quality control. Of the remaining reads on average 37.5% (SD 6.4%) mapped to the human genome (Supplementary Data 6). Compared to the results in stages 1 and 2, we observed an increased heterogeneity of covered RNA classes. The proportion of reads mapping to miRNAs varied between 0.02 and 5.9% with an average of 0.85% (SD 1.1%; Fig. 4c). Most reads mapped to non-annotated intergenic regions, protein-coding genes, ribosomal RNAs (rRNAs), lncRNAs, as well as to transfer RNAs (from GtRNAdb) and snoRNAs. As expected, in our negative controls, nearly no reads mapping to miRNAs, tRNAs, or snoRNAs were found (Supplementary Data 6). Next, we investigated the miRNA read duplication rates and found that the number of reads per UMI was consistent for most cells between the patients with on average 4.74 reads per UMI (SD 2.16; Fig. 4d). Subsequently, we evaluated the number of miRNA molecules found per cell. On average, 389.49 (SD 496.43) molecules per patient cell could be detected, only 7 and 14 molecules were detected in the two negative samples (Supplementary Fig. 11). We thus excluded cells that were likely of low quality by requiring at least 50 detected miRNA molecules, since we expected these to be unlikely to contain spurious signals. Among the remaining 53 cells, the most abundant miRNAs were miR-375-3p, miR-26a-5p, and let-7a-5p (Fig. 4e). Per single cell, we detected a median of 68 miRNAs, with ten miRNAs expressed in over 90% of tested cells (Supplementary Fig. 12). Altogether, all cells expressed 352 unique miRNAs (Supplementary Data 7). The six most variable miRNAs were miR-100-5p, miR-10b-5p, miR-182-5p, miR-200b-3p, miR-335p-3p, and miR-7-5p (Fig. 4f). We computed a UMAP embedding of the cells and clustered them with the Louvain community detection algorithm[23] into six clusters to investigate the miRNA expression variability between patients and between cells of the same patient. As highlighted in Fig. 4g, the cells only moderately clustered per patient (no patient formed its own cluster) indicating a high variability between cells of the same patient, which was underlined by adjusted mutual information of 0.31. Because the number of CTCs per patient is very variable (Fig. 4b), we repeated the clustering analysis for only the three patients with the largest number of sequenced CTCs and we still do not observe clustering by the patient (Fig. 4h).

**Known and non-annotated miRNAs highlight relevance for cancer**. To understand the relevance of miRNAs in CTCs, we performed a pathway enrichment analysis. The miRNAs were sorted by decreasing expression in the CTCs and processed using the gene set enrichment analysis (GSEA) of miEAA[22]. In terms of

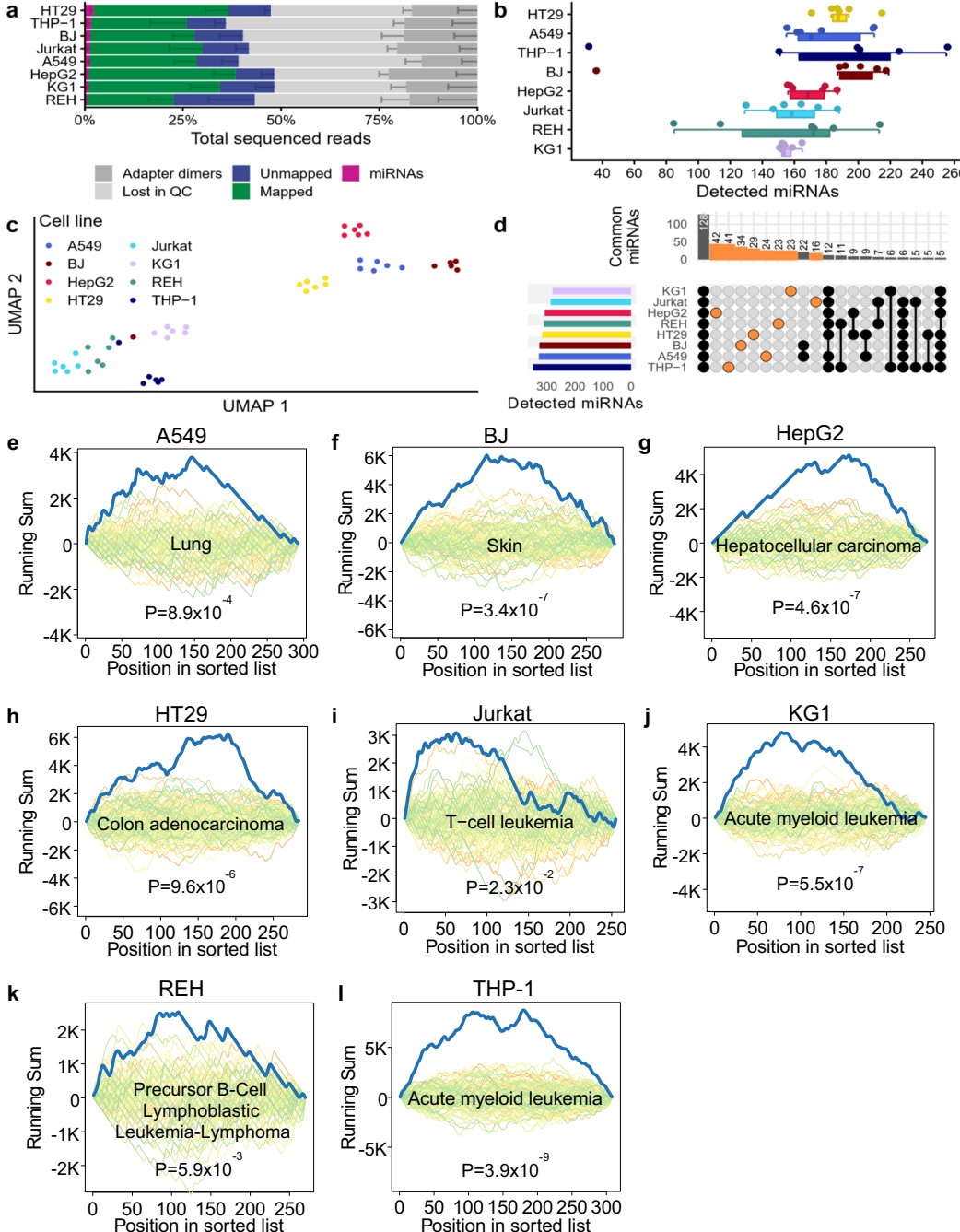

**Fig. 3 Application of protocol SBN_CL to eight different cell lines. a** Read distribution for all tested cell lines, sorted by miRNA reads proportion. The data were presented as mean values ± the standard deviation ($n = 6$ biologically independent samples), which is shown as a smaller error bar in a darker color than its corresponding read group and only represented in one direction. **b** Detected miRNAs for every sample, sorted by decreasing average per cell line shown as boxplot (bottom) and dot plot (top). Each sample is shown as one dot and colored by protocol. Each dot represents one sample. The boxes span the first to the third quartile with the vertical line inside the box representing the median value. The whiskers show the minimum and maximum values or values up to 1.5 times the interquartile range below or above the first or third quartile if outliers are present. **c** UMAP embedding of all samples. Each dot represents one sample. **d** Upset plot showing the miRNAs jointly detected in multiple cell lines, or exclusively found in only one cell line (orange). The bar plot at the top shows on the y-axis the number of miRNAs detected in the cell lines highlighted by connected black or orange dots in the grid below. The bar plot on the left shows on the x-axis the total number of miRNAs detected in at least one of the samples of the cell line shown on the y-axis. **e–l** Examples of significantly enriched categories for each of the analyzed cell lines. Each plot shows the computed running sum (blue), running sums of random permutations (background), and the FDR adjusted *P* value for the cell line specified in the title. Exact *p* values were computed by the gene set enrichment analysis implementation of miEAA for each enrichment and FDR adjusted, separately for each database. Source data are provided in the Source Data file.

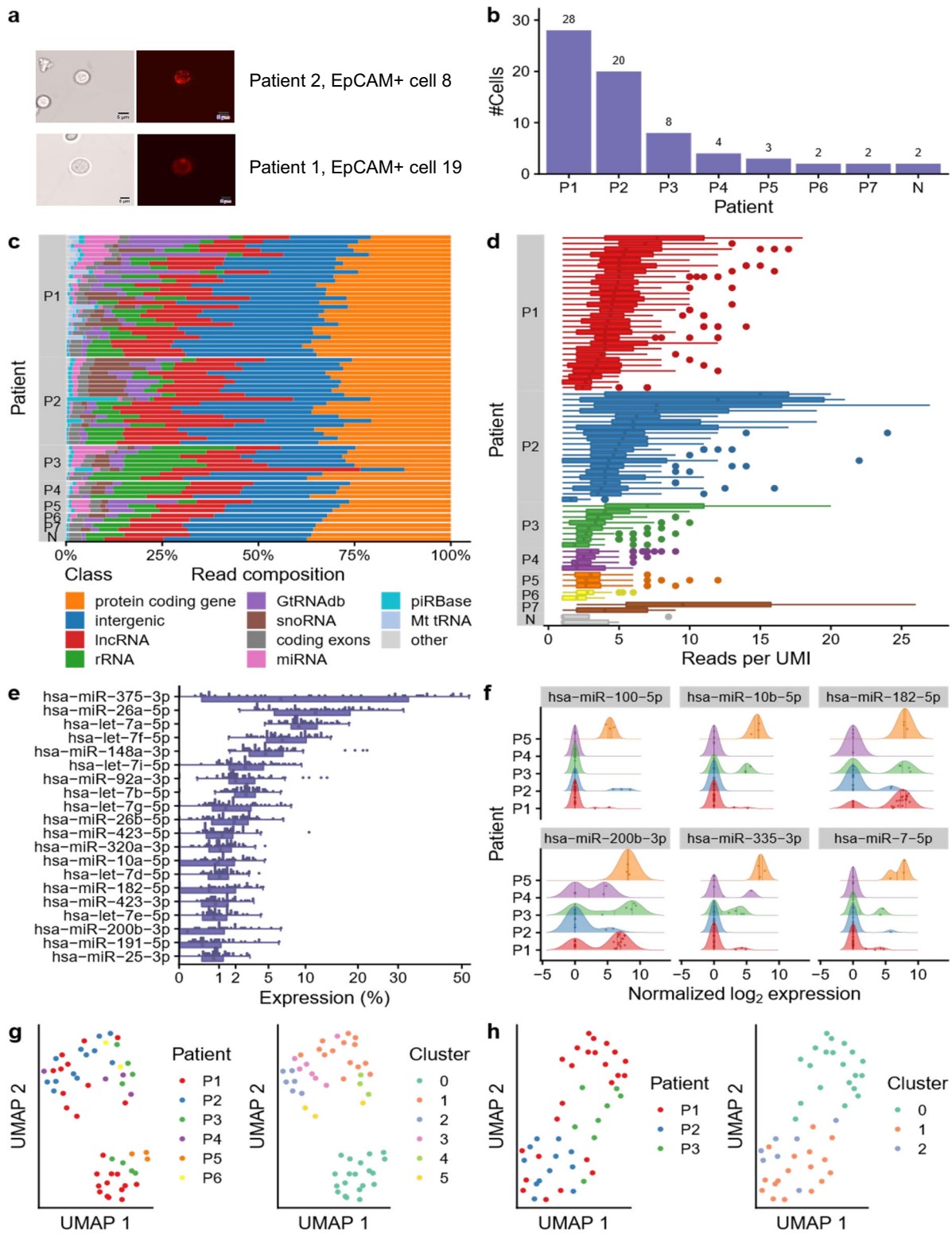

organs from the human miRNA tissue atlas[24], miEAA proposed enrichment in the lung (GSEA, adjusted $p$ value $= 1.5 \times 10^{-11}$) and the colon (GSEA, adjusted $p$ value $= 2.5 \times 10^{-11}$; Fig. 5a, b). Interestingly, the results also suggest an enrichment of several cancer related pathways, such as the integrin signaling pathway (Supplementary Data 8) and cancer as disease including the miR-Walk categories[25] neoplasms (GSEA, adjusted $p$ value $= 4.6 \times 10^{-8}$)

and carcinoma (GSEA, adjusted $p$ value $= 1.6 \times 10^{-6}$; Fig. 5c, d). The most overrepresented cellular localization is the cytoplasm (GSEA, adjusted $p$ value $= 1.6 \times 10^{-15}$) and the mitochondrion (GSEA, adjusted $p$ value $= 1.1 \times 10^{-14}$; Fig. 5e, f). While these results are no functional proof of potential downstream cascades triggered by miRNAs in CTCs, we can at least claim a hypothetic regulatory effect of miRNAs in CTCs.

**Fig. 4 Application of protocol SBN_CL to CTCs of small cell lung cancer (SCLC) patients. a** Representative images of two EpCAM+ cells enriched from blood samples of two SCLC patients (20x). The cells were isolated by micromanipulation for miRNA sequencing. The scale bar is equivalent to 5 μm. **b** Number of cells sequenced per patient and the two empty negative controls. **c** Distribution of the mapped reads for each cell grouped per patient and ordered by descending miRNA proportion. **d** Boxplot showing the number of reads per UMI for each cell, grouped by the patient in descending order. Each miRNA is shown as one dot. The boxes span the first to the third quartile with the vertical line inside the box representing the median value. The whiskers show the minimum and maximum values or values up to 1.5 times the interquartile range below or above the first or third quartile if outliers are present. **e** Distribution of the top 20 most expressed miRNAs across all cells shown as boxplot (bottom) and dot plot (top). The boxes span the first to the third quartile with the vertical line inside the box representing the median value. The whiskers show the minimum and maximum values or values up to 1.5 times the interquartile range below or above the first or third quartile if outliers are present. **f** Expression distribution of the six most variable miRNAs. Patients with less than three cells were excluded. The vertical lines inside the areas delimit the quartiles. The dots inside the area represent the expression of a miRNA in the cell of the corresponding patient. **g** UMAP embedding of all samples colored by the patient (left) and colored by cluster (right). **h** UMAP embedding of the three patients with the highest number of CTCs colored by the patient (left) and colored by cluster (right). Source data are provided in the Source Data file.

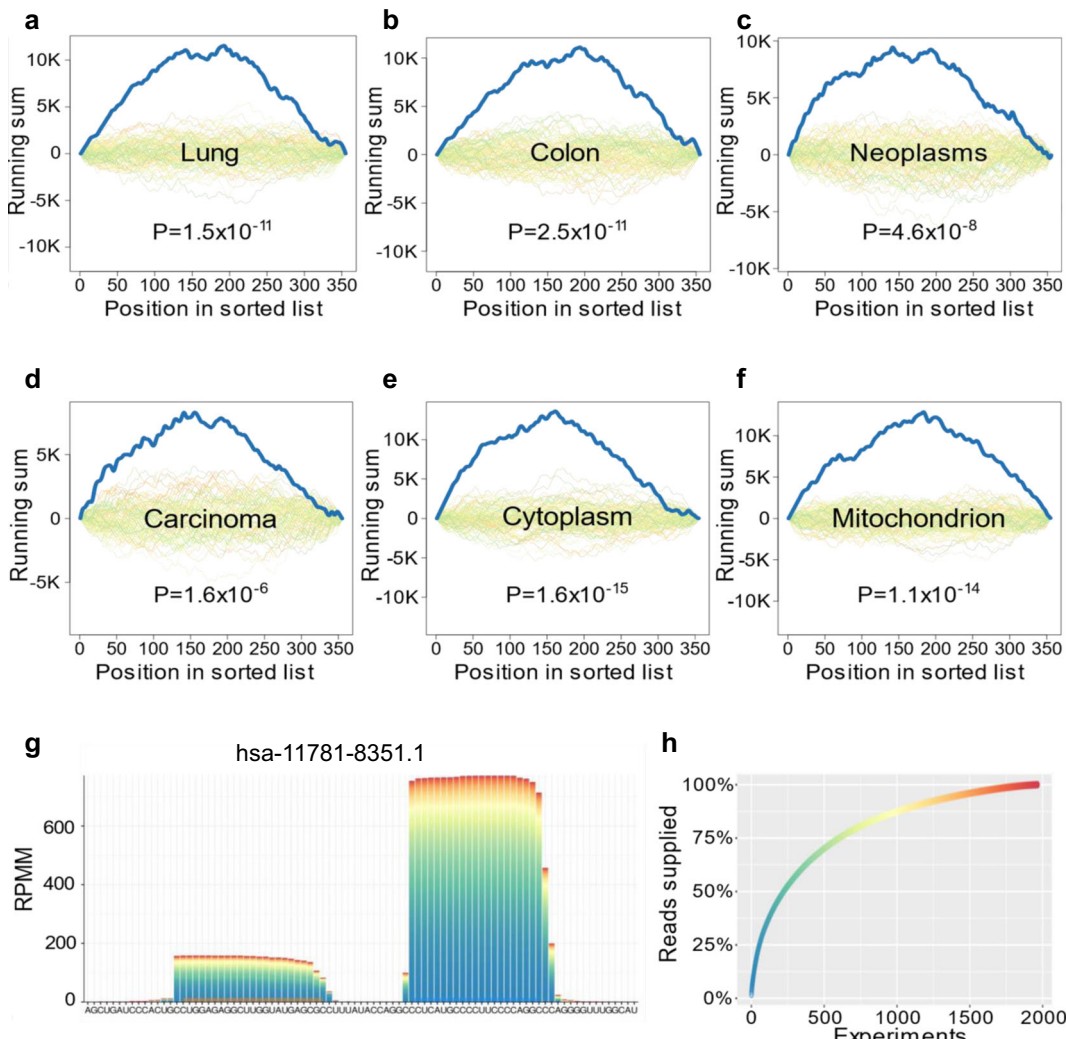

**Fig. 5 Enrichment analysis of CTC miRNAs and miRNA candidates. a–f** Top enriched organs, pathway categories, and cellular locations. Each plot shows the computed running sum (blue), running sums of random permutations (background), and the FDR adjusted *P* value. Exact *p* values were computed by the gene set enrichment analysis implementation of miEAA for each enrichment and FDR adjusted, separately for each database. **g** Pileup plot obtained from miRCarta for one of the overlapping miRNA candidates. The bars are colored according to the experiments the reads were contributed from. The expression is shown as reads per million mapped (RPMM). **h** Distribution of the read proportion supplied by each experiment that detected this miRNA.

The analyses performed so far were restricted to miRNAs annotated in the miRBase. MiRNAs that have not yet been reported or at least miRNAs that have not yet been added to the miRBase may have also regulatory effects. We thus performed a prediction of non-annotated miRNAs in the single CTCs. Our analysis suggested ten non-annotated miRNA candidates. Since the coverage of respective candidates is limited on the single cell level, we set to compute the coverage in bulk sequencing data. Indeed, in four cases we found hits for the potential candidates in miRCarta[26], namely hsa-11781-8351.1, hsa-2644-2657.1, hsa-2810-2791.1, and hsa-9809-4031.1, partly with excellent read mapping profiles (Fig. 5g, h; Supplementary Fig. 13).

Interestingly, the discovered miRNAs that were supported by data from miRCarta also regulated relevant pathways, including cell junction (GSEA, adjusted $p$ value of $4 \times 10^{-8}$) and cell adhesion (GSEA, adjusted $p$ value of $5 \times 10^{-6}$).

## Discussion

In this study, we comprehensively compared 18 ligation-based miRNA-Seq and one polyadenylation-based protocol. The only polyadenylation-based protocol investigated, showed in the first stage lower performance and was excluded together with 10 of the 18 ligation-based protocols. These results are in line with similar protocol comparisons on the bulk level[8,10,11,13]. Eight different protocol variants showed promising results for miRNA-Seq from very low input samples. Both, exonuclease digestion of excess adapters, and chemically modified adapters led to a reduction of adapter dimer formation, with CleanTag adapters[20] appearing to be the most effective strategy. Surprisingly, adapters with random nucleotides at the ligation sites do not show improved miRNA quantification accuracy of the equimolar spike-in miRNAs, and in addition, 4N libraries contained more than 40% of adapter dimer reads. Since similar miRNAs are under- or overrepresented between the different protocols, we showed the observed bias seems to be partially caused by miRNA sequence G-content. Other studies suggest the nucleotide sequence at 5′ or 3′ end, miRNA free energy, enthalpy, entropy, and secondary structure formation as additional causes of bias[9,15].

Single cells probably contain much less than 1 pg miRNA as deduced from the much lower performance of all tested protocols in stage 2 compared to stage 1. Surprisingly, the original SB protocol showed in our experiments an increase in performance compared to the updated SBN protocol. More miRNAs could be detected, and the amount of adapter dimers was lower, which might be explained by the higher adapter concentrations of the SBN protocol. The SB and the SBN_CL protocol can be recommended for the use on single cell level, with advantages in terms of reproducibility for the SBN_CL protocol. We demonstrated the applicability of the SBN_CL protocol in a broad range of cell types. However, this protocol could also be further improved based on our insights, e.g., to detect even more cell-type specific (novel) miRNAs, to investigate the distribution of 3′ and 5′ miRNA arm usage, and to analyze isomiR expression in health and disease[27,28]. Future research should focus on the increase of the number of reads mapping to miRNAs and the reduction of adapter dimers. This might be achieved by optimizing ligation reactions with special attention to the adapter concentrations[29], by the usage of splint adapters[12], by adapter-miRNA-circularization[10,30], by application of CRISPR/Cas9 to deplete adapter dimer reads[16], or by a combination thereof. Furthermore, combined profiling of single cell mRNA and miRNA expression seems possible[31,32]. The current SBN_CL protocol allows sc-miRNA-Seq of about 15 samples within 2 days for library preparation. The protocol could also be easily automated in 96- or 384-well format due to bottom-up reactions and the avoidance of gel or column-based purification steps. Further increase in throughput might be achieved by introducing a barcode to the 3′ adapter and pooling multiple samples after ligation[29].

Our pilot study on seven SCLC patients demonstrates the feasibility of single cell miRNA profiles as potential biomarkers. We have identified many different oncogenic miRNAs in SCLC CTCs. The most abundant miRNAs from the CTC study are known as cancer miRNAs. A comprehensive literature review revealed that miR-21-5p, miR-146b-5p, miR-142-5p, miR-148a-3p, miR-92a-3p, miR-26a-5p, and miR-25-3p were each found to be connected to cancer in over 50 manuscripts. Of note, from all 30 miRNAs that were present in at least 60% of CTCs, 27 have

already been suggested to be of relevance for lung cancer, with the most prominent cases of miR-21-5p[2,33] and miR-142-5p[34,35]. Until now, changes in expression profiles of these miRNAs have only been associated with clinical and molecular features of non-small cell lung cancer primary tumors and circulating miRNA[36], but not with CTCs. Our study shows that these miRNAs are also frequently expressed in CTCs of SCLC patients. Pathway enrichment also provided a first glimpse into the biology of CTCs, as the integrin signaling pathway was the top enriched pathway in SCLC CTCs. Integrin expression seems to be directly related to the aggressiveness of SCLC comprising high metastatic potential and resistance development to chemotherapy[37–41]. To explore the biological relevance and the diagnostic potential of CTC-derived miRNAs, however, larger cohorts with more cases and controls, repeated sampling over time, outcome data, and mechanistic studies are required. Our comprehensive protocol comparison providing an assay for measuring miRNAs in CTCs of cancer patients paves the way for this goal.

## Methods

**Cell culture.** In this study the following cell lines and culture media were used: The breast cancer cell line MCF7 (ATCC HTB22) was cultivated in RPMI medium supplemented with 10% fetal bovine serum, 100 µg ml⁻¹ PenStrep, 0.01 mg ml⁻¹ human recombinant insulin (all PAN-Biotech), and 1x GlutaMAX (Life Technologies); the lung cancer cell line A549 (ATCC CCL-185) was cultivated in DMEM medium supplemented with 10% fetal bovine serum, 100 µg ml⁻¹ PenStrep, and 1x GlutaMAX; the fibroblast cell line BJ (ATCC CRL-2522) was cultivated in MEM medium supplemented with 10% fetal bovine serum, 100 µg ml⁻¹ PenStrep, and 1x GlutaMAX; the hepatocellular carcinoma cell line HepG2 (ATCC HB-8065) was cultivated in MEM medium supplemented with 10% fetal bovine serum, 100 µg ml⁻¹ PenStrep, and 1x GlutaMAX; the colorectal adenocarcinoma cell line HT29 (ATCC HTB-38) was cultivated in RPMI medium supplemented with 10% fetal bovine serum, 100 µg ml⁻¹ PenStrep, 1 mM pyruvate (all PAN-Biotech), and 1x GlutaMAX; the acute leukemia T-cell line Jurkat (ATCC TIB-152) was cultivated in RPMI medium supplemented with 10% fetal bovine serum, 100 µg ml⁻¹ PenStrep, and 1x GlutaMAX; the acute leukemia macrophage cell line KG1 (ATCC CCL-246) was cultivated in IMDM medium supplemented with 20% fetal bovine serum, 100 µg ml⁻¹ PenStrep (all PAN-Biotech), and 1x GlutaMAX; the acute leukemia lymphoblast cell line REH (ATCC CRL-8286) was cultivated in RPMI medium supplemented with 10% fetal bovine serum, 100 µg ml⁻¹ PenStrep, and 1x GlutaMAX; and the acute leukemia monocyte cell line THP-1 (ATCC TIB-202) was cultivated in RPMI medium supplemented with 10% fetal bovine serum, 100 µg ml⁻¹ PenStrep, 1 mM pyruvate, and 1x GlutaMAX. Cell cultures were incubated at 37 °C and 5% $CO_2$. Adherent cell lines were passaged at 80% confluence using 1x Trypsin (PAN-Biotech) every 3–4 days. Non-adherent cells were splitted 1:3 twice a week.

**Cell isolation.** Cells of passage 3–12 were washed with PBS (Life Technologies), centrifuged down (300×g for 5 min), resuspended in PBS, and placed on Adcell™ diagnostic slides (Thermo Fisher Scientific). Single cells were isolated under the microscope in 1 µl PBS using a micromanipulator (Patchman NP2) with pump (CellTram, both Eppendorf) and placed into 2 µl of lysis buffer (0.2% Triton X-100 [Sigma-Aldrich] and 4 U recombinant RNase inhibitor [Clontech Takara]). Samples were stored at −80 °C for up to 6 months.

**miRNA library preparation.** All pipetting steps were conducted on ice; master mixes were added to the edge of the PCR tube and centrifuged down unless otherwise stated. The sequences of the oligonucleotides used are listed in Supplementary Table 4.

*Spike-in experiments.* For the spike-in experiments (stage 1), initially, three replicates were performed. Three additional replicates were conducted for the best-performing protocols. One microliter of the miRXplore Universal Reference (Miltenyi) diluted to 0.96 pg µl⁻¹ (i.e., 1 fg of every spike-in miRNA) was added to every single MCF7 cell isolated in lysis buffer. In order to mask 5.8 S rRNA, 1 µl of a 10 µM blocking oligonucleotide (Metabion) was added, the samples were incubated at 72 °C for 20 min, and immediately cooled on ice.

*Single cell experiments.* After thawing isolated single cells, 2 µl of a 5 µM 5.8 S rRNA blocking oligonucleotide were added to every sample. Next, incubation at 72 °C for 20 min, followed by cooling on ice, was performed. To produce single cell equivalents of the MCF7 cell line (stage 2), the volume (5 µl each) of all reactions was pooled, mixed thoroughly, and redistributed into fresh 0.2 ml PCR tubes. In total, six replicates of every protocol variant in stage 2 were processed.

*Sandberg protocol I (SB).* The applied protocol is based on Faridani et al.[18]. Two microliters of the 3′ adapter ligation master mix (17 nM 3′ adapter, 8% PEG 8000, 50 U T4 RNA ligase 2 truncated KQ, 0.7x T4 RNA ligase buffer [all NEB], and 4 U recombinant RNase inhibitor [Clontech Takara]) were added, the samples were incubated at 30 °C for 6 h and then at 4 °C for 10 h. Removal of the unligated 3′ adapter was performed using 2.5 µl of the following master mix: 200 nM reverse transcription primer, 2.5 U Lambda Exonuclease and 10 U 5′-Deadenylase (both NEB). The samples were incubated at 30 °C, followed by 37 °C for 15 min, both. Then, the 5′ adapter was ligated using a 1.5 µl master mix: 90 nM 5′ adapter, 0.64 mM Tris-buffered ATP, 4 U T4 RNA ligase (both Thermo Fisher), and 0.23x T4 RNA ligase buffer (NEB). Incubation for 1 h at 37 °C was performed. For reverse transcription, 7 µl master mix (1.28x Taq DNA polymerase buffer, 0.42 mM dNTPs [both Roche], 8.33 mM DTT, 150 U Super Script II reverse transcriptase [both Thermo Fisher], and 4 U recombinant RNase inhibitor [Clontech Takara]) were added and the samples were incubated at 42 °C for 1 h. Next, the cDNA was amplified using 32 µl of the following PCR master mix: 2 µM RP1 primer, 0.13 mM dNTPs (Roche), 1 U Phusion Hot Start II DNA polymerase, and 1x Phusion buffer (both Thermo Fisher) and the cycler program: 98 °C for 30 s, 13 cycles of 98 °C for 10 s, 60 °C for 30 s, 72 °C for 30 s, and a final incubation at 72 °C for 5 min. In a second PCR amplification with a total volume of 25 µl (200 nM RP1 primer, 0.2 mM dNTPs [Roche], 0.5 U Phusion Hot Start II DNA polymerase, and 1x Phusion buffer [both Thermo Fisher]), 1 µl of the first PCR was used as a template and 2 µM sample-specific indexing primer was added. A similar cycler program as for the first PCR was used, but the annealing temperature was 67 °C. Finally, an Ampure XP bead (Beckman Coulter) size selection was performed. Samples were mixed with beads in a ratio of 1:1, incubated for 10 min at RT, incubated for 4 min on a magnet, the supernatant was transferred to a fresh tube and mixed with Ampure XP beads in a ratio of 1:1.6. After incubation, the supernatant was discarded, the beads were washed twice with 80% ethanol and the sample was eluted in 15 µl water. Samples can be stored at −20 °C.

*Protocol variants tested.* SB_4N: The 3′ and 5′ adapters were replaced by adapters with four random nucleotides at the ligation sites[9].

SB_CL: The 3′ and 5′ adapters were replaced by the adapters of the modified CleanTag protocol[20].

SB_C3: The 3′ adapter was exchanged by the 3′ CleanTag adapter[20] and the 5′ adapter was exchanged by an adapter with five 3′ nucleotides complementary to the 3′ adapter[15] and the UMI was shortened to 6 nt.

*Sandberg protocol II (SBN).* The SBN protocol[19] is an optimized version of the SB protocol[18]. Three microliters of the 3′ adapter ligation master mix (2 µM 3′ adapter, 8% PEG 8000, 50 U T4 RNA ligase 2 truncated KQ, 0.8x T4 RNA ligase buffer [all NEB], and 4 U recombinant RNase inhibitor [Clontech Takara]) were added, the samples were incubated at 30 °C for 6 h and then at 4 °C for 10 h. Removal of the unligated 3′ adapter was performed using 2 µl the following master mix: 5 µM reverse transcription primer, 2.5 U Lambda Exonuclease and 25 U 5′-Deadenylase (both NEB). The samples were incubated at 30 °C, followed by 37 °C for 15 min. Then, the 5′ adapter was ligated using a 2 µl master mix: 1 µM 5′ adapter, 0.67 mM Tris-buffered ATP (Thermo Fisher), 13.5 U T4 RNA ligase 1 ssRNA and 0.25x T4 RNA ligase buffer (both NEB). Incubation for 1 h at 37 °C was performed. For reverse transcription, 5 µl master mix (1.3x Taq DNA polymerase buffer, 0.5 mM dNTPs [both Roche], 8 mM DTT, 100 U Super Script II reverse transcriptase [both Thermo Fisher], and 4 U recombinant RNase inhibitor [Clontech Takara]) were added and the samples were incubated at 42 °C for 1 h, followed by 70 °C for 15 min. Next, the cDNA was amplified using 13 µl of the following PCR master mix: 1 µM RP1 primer, 0.15 mM dNTPs (Roche), 1 U Phusion Hot Start II DNA polymerase, and 1x Phusion buffer (both Thermo Fisher) and the cycler program: 98 °C for 30 s, 13 cycles of 98 °C for 10 s, 60 °C for 30 s, 72 °C for 30 s, and a final incubation at 72 °C for 5 min. In a second PCR amplification with a total volume of 25 µl (800 nM RP1 primer, 0.2 mM dNTPs [Roche], 0.5 U Phusion Hot Start II DNA polymerase, and 1x Phusion buffer [both Thermo Fisher]) 1 ul of the first PCR was used as a template and 2 µM sample-specific indexing primer were added. A similar cycler program as for the first PCR was used, but the annealing temperature was 67 °C. The Ampure XP bead size selection was performed as described for the SB protocol.

*Protocol variants tested.* SBN_4N: The 3′ and 5′ adapters were replaced by adapters with four random nucleotides at the ligation sites[9].

SBN_CL: The 3′ and 5′ adapters were replaced by the adapters of the modified CleanTag protocol[20]. This protocol variant was used in stage 3 to process the single cell line cells and the EpCAM positive cells of the SCLC patients.

*CleanTag protocol (CL).* The protocol was adapted from Shore et al.[20]. An 8 nt UMI was introduced into the published 5′ adapter analogously to the Sandberg protocol[18]. For 3′ CL adapter ligation, 6 µl master mix (26 nM 3′ CL adapter [Biomers], 13% PEG 8000, 200 U T4 RNA ligase 2 truncated KQ, 0.8x T4 RNA ligase buffer [all NEB], and 40 U recombinant RNase inhibitor [Clontech Takara]) were added to the samples and incubation was performed at 28 °C for 1 h and 65 °C for 20 min. Next, the ligation of the 5′ CL adapter was conducted by adding 4.7 µl master mix: 110 nM CL 5′ adapter, 1.8 mM ATP, 10 U T4 RNA ligase, 0.45x T4

RNA ligase buffer (all Thermo Fisher), and 40 U recombinant RNase inhibitor (Clontech Takara). The same incubation conditions as for the 3′ CL adapter ligation were used. Before reverse transcription, 18 nM reverse transcription primer were added, the samples were incubated at 70 °C for 2 min, and immediately placed back on ice. After that, 6.9 µl of the reverse transcription master mix (7.65 mM DTT, 1x first strand buffer, 200 U Super Script II reverse transcriptase [all Thermo Fisher], 0.38 mM dNTPs [Roche], and 40 U recombinant RNase inhibitor [Clontech Takara]) were added and the samples were incubated at 42 °C for 1 h. Next, the cDNA was amplified using 17 µl of the following PCR master mix: 2 µM RP1 primer, 0.2 mM dNTPs (Roche), 1 U Phusion Hot Start II DNA polymerase, and 1x Phusion buffer (both Thermo Fisher) and the cycler program: 98 °C for 30 s, 13 cycles of 98 °C for 10 s, 60 °C for 30 s, 72 °C for 30 s, and a final incubation at 72 °C for 5 min. In a second PCR amplification with a total volume of 25 µl (200 nM RP1 primer, 0.2 mM dNTPs [Roche], 0.5 U Phusion Hot Start II DNA polymerase, and 1x Phusion buffer [both Thermo Fisher]), 1 µl of the first PCR was used as a template and 2 µM sample-specific indexing primer was added. A similar cycler program as for the first PCR was used, but the annealing temperature was 67 °C. The Ampure XP bead size selection was performed as described for the SB protocol.

*Protocol variants tested.* CL_16C: The 3′ adapter ligation reaction was incubated at 16 °C for 15 h followed by 65 °C for 20 min[8].

CL_4N: The 3′ and 5′ adapter were replaced by adapters with four random nucleotides at the ligation sites[9].

CL_Block: The 5′ CL adapter was replaced by an adapter with three additional uridine nucleotides at the 3′ end and an UMI shortened to 6 nt. Additionally, an altered reverse transcription primer was used, which could bind to adapter dimers. Before reverse transcription, 18 nM Block_RT primer and 1 U USER enzyme (NEB) were added. Incubation was performed at 42 °C for 10 min, 37 °C for 15 min, and 65 °C for 10 min to digest adapter dimers.

CL_C3: The 5′ CL adapter was exchanged by an adapter with five 3′ nucleotides complementary to the 3′ adapter[15] and the UMI was shortened to 6 nt.

CL_Rand: The 3′ CL adapter was replaced by an adapter that had the first six 3′ nucleotides exchanged against random nucleotides[15]. In addition, truncated versions of the reverse transcription primer and the PCR indexing primers were used.

CL_SB: The 3′ and 5′ CL adapters were replaced by adapters of the Sandberg protocol[18].

CL_UMI6: The UMI of the 5′ adapter was shortened from 8 nt to 6 nt.

*4N protocol (4N).* The protocol was adapted from the 4N Protocol A of Giraldez et al.[9]. For 3′ 4N adapter ligation, 1.5 µl adapter mix (111 nM 3′ 4N adapter and 5.5% PEG 8000 [NEB]) were added and the samples were incubated at 70 °C for 2 min and immediately placed back on ice. Then, 2.5 µl of the ligation master mix (200 U T4 RNA ligase 2 truncated KQ, 1x T4 RNA ligase buffer [both NEB], and 8 U recombinant RNase inhibitor [Clontech Takara]) were added and the samples were incubated at 25 °C for 2 h. The single-stranded DNA binding protein (Promega) was diluted to 2 µg µl⁻¹ using 1x T4 RNA ligase buffer (NEB), 0.5 µl were added to every sample, and incubation at 25 °C for 10 min was performed. After that, 25 U 5′ deadenylase and 15 U RecJf (both NEB) were added sequentially and incubation at 30 °C for 1 h and 37 °C for 1 h, respectively, was performed. Next, the ligation of the 5′ 4N adapter was conducted: First, the 5′ 4N adapter was heated to 70 °C for 2 min, then 2 µl of master mix (80 nM 4N 5′ adapter, 400 µM ATP [Thermo Fisher], and 10 U T4 RNA ligase 1 ssRNA [NEB]) were pipetted to every sample, and finally, incubation was performed at 25 °C for 1 h. For reverse transcription, 6.5 µl master mix (53 nM reverse transcription primer, 0.26 mM dNTPs [Roche], 0.8x first strand buffer, 5.26 mM DTT, and 200 U Super Script III reverse transcriptase [all Thermo Fisher]) were added and the samples were incubated at 55 °C for 1 h, followed by 37 °C for 15 min. Next, the cDNA was amplified using 30 µl of the following PCR master mix: 2 µM RP1 primer and 1x NEBNext Ultra II Q5 master mix (NEB). Additionally, 2 µM of an individual indexing primer were added to every sample. The following cycler program was applied: 98 °C for 30 s, 20 cycles of 98 °C for 10 s, 60 °C for 30 s, 65 °C for 35 s, and a final incubation at 65 °C for 2 min. The Ampure XP bead size selection was performed as described above for the SB protocol.

*Protocol variants tested.* 4N_C3: The 3′ adapter was exchanged by the 3′ CleanTag adapter[20] and the 5′ adapter was exchanged by an adapter with five 3′ nucleotides complementary to the 3′ adapter[15] and the UMI was shortened to 6 nt.

4N_CL: The 3′ and 5′ adapters were replaced by the adapters of the modified CleanTag protocol[20].

*CATS protocol (CATS).* The CATS small RNA sequencing kit for Illumina from Diagenode was used, which is ligation-free and based on polyA tailing of miRNAs[21]. The following changes were applied to adapt the protocol for single cell input: cell lysis and removal of 5.8 S rRNA were performed as described above, the number of PCR cycles was increased to 24 and an Ampure XP bead size selection instead of gel-based size selection was performed.

*Quality controls.* The concentration of every sample was determined using the Qubit High Sensitivity DNA kit (Life Technologies). If necessary, the samples were

diluted to 1.8 ng µl⁻¹ and the fragment length distribution was evaluated on a Bioanalyzer High Sensitivity dsDNA chip (Agilent Technologies). Adapter dimers are expected to show fragment lengths of around 125 bp, whereas libraries with miRNA insert should show fragment lengths of around 145 bp.

*Illumina sequencing*. The libraries were quantified using the KAPA Library Quantification Kit for Illumina (Roche). For one MiSeq run, 24–40 libraries were pooled in equimolar concentration. A final concentration of 18–20 pM library spiked with 5% PhiX (Illumina) was sequenced (75 bp, single read) using a MiSeq Reagent Kit V3—150 Cycles (Illumina). Raw sequencing data is freely accessible under the project identifier SRP279094 from the Sequence Read Archive.

**Small cell lung cancer patients**. For CTC quantification and isolation, whole blood was drawn from seven SCLC patients (ethic vote 07-079, approved by the University of Regensburg Ethics Committee). Patients with SCLC were recruited at two local therapy centers (University Hospital Regensburg and Krankenhaus Barmherzige Brueder Regensburg). Inclusion criteria were histologically confirmed SCLC Stage IV, ability to understand and sign an informed consent form. No other inclusion criteria were defined. Exclusion criteria were inability to follow the study protocol and inability to give written informed consent. Informed written consent was obtained from all participating patients. The patient data is summarized in Supplementary Table 5.

**Blood processing and EpCAM staining**. About 7.5 ml of blood was collected in CellSave tubes for CTC enumeration using the CellSearch System (Menarini Silicon Biosystems). For miRNA-Seq, CTCs were enriched from 10 ml EDTA blood by density gradient centrifugation with 50% Percoll (GE Healthcare) at 1000xg for 20 min. Cells from interphase were either cryopreserved or directly stained with EpCAM-PE antibody (HEA-125, Miltenyi Biotec). Fluorescently labeled single cells were isolated by micromanipulation as described above.

**Bioinformatics**. Illumina BCL files were converted to FASTQ using bcl2fastq (2.20.0.422). In a first step, if a library had a UMI, the UMI of each read was copied in its header. Then adapter trimming and quality trimming were performed with cutadapt 2.10[42]. Reads were trimmed at the 3′ end with a quality cutoff of 20. In addition to the 3′ adapter, polyA tails, polyG tails, and PCR primers were removed. Trimming was performed up to three times per read while allowing an error rate of 10%, at most 2 N bases and requiring a minimum length of 18 bases to keep the read. Adapter dimers were determined as all reads with less than four bases remaining after trimming. The remaining reads were mapped against the primary assembly of the human genome GRCh38 using STAR[43] (2.7.5b) with parameters: "–outFilterMultimapNmax 50–outFilterScoreMinOverLread 0–outFilterMultimapScoreRange 0–outFilterMatchNmin 18–outFilterMatchNminOverLread 0–outFilterMismatchNoverLmax 0.04–alignIntronMax 1–outSAMstrandField intronMotif". RNA classes were identified using featureCounts (1.5.2)[44] with parameters: "-F SAF –O –M –R -f –fracOverlap 0.9", requiring an overlap of at least 90% of reads with annotated regions and allowing multimapping reads and overlapping features. Annotations were extracted from GENCODE v25[45], miRBase v22, piRBase v1[46], and GtRNAdb (18.1)[47]. All regions that were not annotated were tagged as "intergenic". After running featureCounts, we assigned the reads to each category in a hierarchical manner. If a read mapped to more than one category, we assigned it to only one category by prioritizing the classes in the following order: miRNA, miRNA primary transcript, GtRNAdb, Mt tRNA, rRNA, Mt rRNA, snoRNA, snRNA, sRNA, scaRNA, scRNA, piRBase, misc RNA, ribozyme, coding exons, lncRNA, ncRNA, and protein-coding gene. For the samples of the first stage an additional mapping against the miRXplore sequences was performed with RazerS 3 (3.5.3)[48] with parameters: "–forward -so 1 -dr 0 -rr 100–percent-identity 80". The minimum free energy of the miRXplore sequences was computed using RNAfold 2.4.17 with default parameters[49]. The expression of miRNAs was quantified with miRMaster (1.1)[50] and miRNAs were counted as detected if they had a count of at least 1. Read deduplication was performed with UMI-tools (1.0.0)[51] and the adjacency method with the following parameters "–read-length –per-contig –per-gene –random-seed 42 –method adjacency". Sequencing data of stage 1 and 2 were analyzed with and without subsampling. Subsampling was performed with seqtk with the number of reads set to 300,000 and a seed of 42. Plots were generated in R with ggplot2 (3.3.2)[52], gghalves (0.0.1.9000), ggridges (0.5.1), and ComplexUpset (1.2.0). The six most variable miRNAs were determined by the variance stabilizing transformation implemented in Seurat (3.1.1)[53]. The 2D UMAP embedding was created with uwot (0.1.4) with nine neighbors for stage 1, 12 neighbors for stage 2, 12 neighbors for the cell line samples, and 15 neighbors for the CTCs. For the second stage, the cell lines and CTCs, Seurat was used to process the expression matrix by normalizing the reads to logarithmized counts normalized by the library size with a scaling factor of 10,000, determining the variable features using a variance stabilizing transformation ("vst" method), followed by PCA on those features and UMAP on the first 20, respectively 15 PCs. Louvain clustering of the CTCs was performed on a shared nearest neighbor graph based on the first 15 PCs with a resolution of 1.4, resulting in six clusters (the number of patients remaining after QC), as well as for the subset of three patients with most cells with the first 15 PCs and a resolution of 1.05. Enrichment analysis was performed with miEAA 2.0 with default parameters and Benjamini–Hochberg adjustment using all expressed miRNAs ordered

by their average expression. Non-annotated miRNA candidates were predicted with miRMaster (1.1) and matches to miRCarta (v1.1) were determined using the provided upload functionality.

**Reporting Summary**. Further information on research design is available in the Nature Research Reporting Summary linked to this article.

## Data availability
All sequencing raw data generated in this study is freely accessible from SRA with accession SRP279094. The processed data are available at GEO with accession GSE162514. All the other data supporting the findings of this study are available within the article and its Supplementary information files. A reporting summary for this article is available as a Supplementary information file. Source data are provided with this paper.

## Code availability
The scripts and data used to generate the results are available at https://github.com/CCB-SB/sc_mirna_seq_manuscript.

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

## Acknowledgements

We thank Judith Proske and Dominik Wermers for their excellent technical assistance. This study was supported by DFG KL-1233/10-2 and by the Bavarian ministry of economic affairs, energy and technology ("Neue Technologien, Vernetzungen und Forschungsansätze für die personalisierte Medizin", AZ 20-3410.1-1-2) (to C.A.K.).

## Author contributions

Conception and design: S.K. and A.K. Development of methodology: S.M.H., T.F., A.K., and S.K. Acquisition of experimental data: S.M.H. Analysis and interpretation: S.M.H., T.F., A.K., and S.K. Writing of the manuscript: S.M.H., T.F., A.K., and S.K. Acquisition of clinical data: F.L. and A.S.-L. Clinical sample logistics: C.W. and C.A.K. Review and/or revision of the manuscript: all authors. Study supervision: S.K. and A.K.

## Funding

## Competing interests

A.K. and T.F. report a research grant and personal fees from the company Diagenode, manufacturing one of the kits considered in the study. The relation is independent of the current research manuscript and Diagenode is not involved in any way in generating the study or interpreting the data. From May 2021 on, A.K. reports personal fees from the company Firalis, working on non-coding RNA diagnosis. The relation is independent of the current research manuscript and Firalis is not involved in any way in generating the study or interpreting the data. The remaining authors declare no competing interests.
