## [Peer Review File · Nature Communications]

Reviewers' Comments:

Reviewer #1:

Remarks to the Author:

Hucker and colleagues evaluated 19 different protocols for their ability to capture microRNA from individual tumor cells. They describe the protocols in sufficient details which likely allows reproduction by other users. They document the obtained results very well and methodically illustrate the individual steps that lead to the selection of their preferred protocol. The best performing protocol was then used to isolate miRNA profiles from circulating tumor cells obtained from patients with small lung cell cancer.

The work seems well performed and constitutes in my opinion a timely and relevant resource for the field of oncology and beyond. Moreover, it has the potential to become a well cited reference protocol. While I have several minor points there are also a few major issues that need to be addressed.

Major points:

- The authors claim that this method is suitable for profiling miRNA expression pattern in different types of cells but the performance of the protocol across different types of cells was not assessed so far. In order to illustrate that the protocol can be used for the indent purpose, it would be necessary to demonstrate that cell type specific miRNA can be captured with the method.
- The authors analyzed 67 cells in total isolated form 7 patients. Of note, 48 of these cells originate form 2 patients. This bias and the limited number of cells make conclusions about intrapatient and interpatient variance quite difficult. Some critical comments should be added.
- The authors conclude a moderate clustering by sex. Out of 6 patients only two are female, one of which contribute with 3 cells. As more than 90% of the female cells come from one patient, it remains unclear if this is a sex or patient driven effect.
- Moreover, the clustering analysis shown in figure 3g is on the edge, i.e. to show a cluster that contains 2 cells is problematic.
- The discussion section is redundant as it repeats only what was said before. Maybe merging it with the results section would be helpful.
- We understand that the protocol SBN-CL performed very well at the detection of spiked miRNA (stage 1) and endogenous miRNA (Stage2). Somewhat surprisingly, there are also protocols such as CL that performed very well in stage 1 and really not well in stage 2. This discrepancy is slightly difficult to understand. At the same time, figure 2a show that CL had much lower sequencing depth in stage 2 compared to stage 1. This leaves some uncertainty if the measured differences reflect systematical or randomly occurring lower performance. At the same time, one should appreciate the efforts done by the authors and I think more extensive evaluations would be beyond the scope of the article. Nonetheless, it would be helpful to discuss that the limited numbers of repeats some protocols might have inadequately excluded. However, I agree with the decision of authors on the selected protocol.

Minor points:

- The authors use micromanipulation to isolate circulating tumor cells but this approach is not described in the methods section.
- The number of EpCAM positive cells per patient varied between 2 and 28 but what volume of blood was sampled for this purpose.
- Lines 94-98 and Supplementary 2: Out of the 19 methods that were tested in triplicates, only

one bioanalyzer profile was selected to be displayed. To better judge the individual performance it would be helpful to display all replicates using the overlay function of the BioAnalyzer Expert software.

- Lines 120-126 and the related figure 1d-h are difficult to understand. Some rewriting is needed to understand the content and the relevance.

- Lines 142-144: the statement "Remarkably, only ligation-based approaches were among the list of top-performing protocols such that no polyadenylation-based approach was contained in the second stage" is difficult to reconcile. It is not very surprising that no polyadenylation-based protocol was included into the second stage when only 1 was evaluated?

- Figure 2a, c: A more clear reference what sequencing stage 1 and 2 means could be helpful.

- Figure 2h: The x-axis label is confusing. Should it not indicate the percent of samples in which the particular miRNA was detected?

- Figure 2i: It is difficult to understand what is shown.

- Methods section: During the density gradient centrifugation a 1200g was used, which is quite high. Is this correct?

Reviewer #2:

Remarks to the Author:

In this paper, the authors benchmark protocols for the detection of micro RNAs in single cells. To this end, they adapt four different ligation based protocols with variations resulting in a total of 18 protocols plus one polyAdenylation based protocol.

In first testing round 1pg of the microRNA reference panel miRExplore was spiked into single cell equivalents from the cell line MCF7 cell-line and then sequenced in triplicates following each of the 19 protocols being evaluated. The main criteria, were the cDNA yield & size ranges from the bioanalyzer, the number of reads mapping to known miRs and the number of spike in miRs detected. Based on this the field is narrowed to 8 protocols that are now evaluated using real single cell data again in 3 replicates, i.e. only 3 cells. The final method comparison is done using patient data with at the most only 28 cells per patient.

Major things:

1. The manuscript could benefit from a better description of the miRNAs detected and in in the case for which the reference panel is available, contrast them with the ones that were not detected. For Readers, like me, who are not familiar with miRExplore it would also be great to outline the basic properties: How many? Relative concentration ...

2. The initial selection is based on the consistency of 3 replicates, whereas each replicate is a single cell equivalent. In my opinion this is too little, the entire evaluation here might hinge on a single small pipetting inconsistency. Furthermore, including more replicates would allow the authors to provide better statistics. For example they could evaluate the impact of all protocol modifications in a generalised linear model or GC content and secondary structure as the authors suggest in the Discussion.

3. The statistics often don't match the conclusions: How can infer bias from a coefficient of variation, bias refers to a consistent under or overestimation due to technical problems, not to an increase in variance. Similarly, spearman's correlation coefficient is not a good measure of reproducibility. The ordering of expression levels of genes and I guess also miRs is often very robust towards deviations.

4. Also due to the low sample sizes, the results remain rather descriptive. For example the authors report a number of miRs that are unique to one replicate and others that were found everywhere. However, the authors do not investigate this observation further. In my opinion it might well be that such a result is expected. This could be modelled by analysing the sampling distribution.
5. At the end a clear summary, maybe as a figure, that shows the different criteria and the performance of the tested methods is lacking.

All in all, I think the sample sizes and the variable design of the different tests is not very informative for a method comparison paper.

REVIEWER COMMENTS

Reviewer #1 (Remarks to the Author):

Hucker and colleagues evaluated 19 different protocols for their ability to capture microRNA from individual tumor cells. They describe the protocols in sufficient details which likely allows reproduction by other users. They document the obtained results very well and methodically illustrate the individual steps that lead to the selection of their preferred protocol. The best performing protocol was then used to isolate miRNA profiles from circulating tumor cells obtained from patients with small lung cell cancer.

The work seems well performed and constitutes in my opinion a timely and relevant resource for the field of oncology and beyond. Moreover, it has the potential to become a well cited reference protocol. While I have several minor points there are also a few major issues that need to be addressed.

We highly value the encouraging general comments to our work. We appreciate the constructive comments below and addressed them in the revised manuscript.

Major points:

- The authors claim that this method is suitable for profiling miRNA expression pattern in different types of cells but the performance of the protocol across different types of cells was not assessed so far. In order to illustrate that the protocol can be used for the indent purpose, it would be necessary to demonstrate that cell type specific miRNA can be captured with the method.

We agree with the reviewer's comment that the detection of cell type-specific miRNA expression patterns in single cells has been not adequately demonstrated in the original manuscript. Therefore, we additionally performed single cell miRNA analysis from a lung cancer cell line, a colorectal cancer cell line, a hepatocellular cancer cell line, a fibroblast cell line and four hematopoietic cell lines. From each of these cell lines we analyzed a total of six replicates. An overlap analysis highlights that in addition to a common set of miRNAs that was found in all cell lines many captured miRNAs could only be found in specific cell lines (Figure 3d). In addition, we analyzed the miRNA profiles of these cell lines and show that they correspond to the expected patterns (Figure 3e-l).

- The authors analyzed 67 cells in total isolated form 7 patients. Of note, 48 of these cells originate form 2 patients. This bias and the limited number of cells make conclusions about inpatient and interpatient variance quite difficult. Some critical comments should be added.

We agree with the reviewer that no general conclusions can be drawn from such a low number of different patients. The high variability in the number of detected CTCs per patient reflects the clinical reality. Figure 4h shows that the heterogeneity of the CTCs miRNA profiles from the three patients with the highest number of detected CTCs is large. Of course, we do not know if SCLC patients in general show a high heterogeneity in the miRNA expression profiles of their CTCs and therefore, we have adapted the title of this chapter (line 248) and the description of the UMAP analysis (lines 282-285).

- The authors conclude a moderate clustering by sex. Out of 6 patients only two are female, one of which contribute with 3 cells. As more than 90% of the female cells come from one patient, it remains unclear if this is a sex or patient driven effect.

We agree with the reviewer and consequently removed this analysis, since a clear attribution is not possible.

- Moreover, the clustering analysis shown in figure 3g is on the edge, i.e. to show a cluster that contains 2 cells is problematic.

This is a very valid comment. We thus refined the cluster analysis and now only show clusters with at least 3 cells. In addition, we also repeated the analysis only for the cells of the three patients with the largest number of cells and show that the clustering is still mixed (Fig. 4g and 4h).

- The discussion section is redundant as it repeats only what was said before. Maybe merging it with the results section would be helpful.

We appreciate this constructive comment. We now added significantly more information to the results section. While we see the value of combining the two sections and while we are aware that a certain repetition can't be avoided in the discussion section, we finally left the two parts separate. The main reason was the length and structure of the combined section that made it hard to follow. To address the concern of the reviewer we however avoided repeated parts in the discussion section and left them only where required to understand the context of the respective part.

- We understand that the protocol SBN-CL performed very well at the detection of spiked miRNA (stage 1) and endogenous miRNA (Stage2). Somewhat surprisingly, there are also protocols such as CL that performed very well in stage 1 and really not well in stage 2. This discrepancy is slightly difficult to understand. At the same time, figure 2a show that CL had much lower sequencing depth in stage 2 compared to stage 1. This leaves some uncertainty if the measured differences reflect systematical or randomly occurring lower performance. At the same time, one should appreciate the efforts done by the authors and I think more extensive evaluations would be beyond the scope of the article. Nonetheless, it would be helpful to discuss that the limited numbers of repeats some protocols might have inadequately excluded. However, I agree with the decision of authors on the selected protocol.

We agree with the reviewer that the number of replicates is a minimum and we also explicitly thank the reviewer for understanding that a significantly higher number of replicates is actually beyond the scope of the study. Nevertheless, as the second reviewer also raised concerns on the performance evaluation of the different protocols due to the limited number of replicates, we doubled the number of replicates in all stages of the protocol evaluation process.

Already in the original manuscript we sequenced and analyzed 148 miRNA libraries (56 in stage 1, 23 in stage 2 and 69 in stage 3). We have now increased this number to 244 miRNA libraries (80 in stage 1, 47 in stage 2 and 117 in stage 3). This includes the additional replicates in the protocol evaluation stages and the experiments to corroborate the detection of cell type-specific miRNA expression patterns.

The additional replicates of stages 1 and 2 confirmed the results of the original three replicates per protocol variant. Because we also noticed the differences in the number of sequenced reads per protocol variant (Fig. 2a), we subsampled all libraries down to 300,000 reads, repeated all analyses and confirmed the results of stage 1 (Supplementary Fig. 7) and stage 2 (Supplementary Fig. 10). Indeed, several protocols show a sufficient performance in stage 1 with miRXplore spike-in, whereas in stage 2 on single cell level their performance was quite poor. We suppose this discrepancy is caused

by the much lower miRNA content of a single cell compared to 1 pg miRXplore spike-in (lines 332-339).

Minor points:

- The authors use micromanipulation to isolate circulating tumor cells but this approach is not described in the methods section.

Thank you for pointing this out. We have revised the chapter cell isolation in the Methods section to provide further details about micromanipulation (lines 403-410).

- The number of EpCAM positive cells per patient varied between 2 and 28 but what volume of blood was sampled for this purpose.

10 mL of whole blood from an EDTA tube was used per patient. This information can be found in the chapter blood processing and EpCAM staining of the Methods section (line 607).

- Lines 94-98 and Supplementary 2: Out of the 19 methods that were tested in triplicates, only one bioanalyzer profile was selected to be displayed. To better judge the individual performance it would be helpful to display all replicates using the overlay function of the BioAnalyzer Expert software.

Thank you for pointing this out. We've now used the Bioanalyzer Expert software to overlay the electropherograms of all replicates of a protocol in one graph each. Overlaid Bioanalyzer profiles are illustrated in Supplementary Figure 2 (Stage 1) and Supplementary Figure 8 (Stage 2).

- Lines 120-126 and the related figure 1d-h are difficult to understand. Some rewriting is needed to understand the content and the relevance.

We rewrote the corresponding sentences which now reads (lines 133-143):

First, we performed a dimension reduction analysis via UMAP which showed that samples cluster according to the used protocol. We observed that samples processed with the 5' and 3' 4N adapters showed a clear split in comparison to the other protocols (**Fig. 1d**). In a next step, we evaluated the reproducibility of the measurements of each sample and found that the SB protocol showed the highest reproducibility (lowest Euclidean distance between the replicates of the same protocol), followed by the SB_4N and SBN_CL protocol, while the 4N protocols (4N, 4N_CL and 4N_C3) showed the lowest reproducibility (**Fig. 1e**). A comparison of the single protocols to all other protocols highlighted that the samples of protocol 4N had the highest Euclidean distance, i.e., were the most different from all other protocols.

- Lines 142-144: the statement "Remarkably, only ligation-based approaches were among the list of top-performing protocols such that no polyadenylation-based approach was contained in the second stage" is difficult to reconcile. It is not very surprising that no polyadenylation-based protocol was included into the second stage when only 1 was evaluated?

We agree with the reviewer that it is not particularly surprising if only 1 of 19 protocols is polyadenylation-based, no polyadenylation protocol makes it to the second round of evaluation. This fact was rephrased to clarify that only 1 polyadenylation protocol for low input miRNA sequencing in total has been published at all to date (167-169).

- Figure 2a, c: A more clear reference what sequencing stage 1 and 2 means could be helpful.

We have revised Figure 1a and the chapter experimental design of the results section to explain in further detail the different stages of experiments (lines 73-99). Additionally, a short description of stage 1 and 2 was added to the legend of Figure 2a (lines 711-713).

- Figure 2h: The x-axis label is confusing. Should it not indicate the percent of samples in which the particular miRNA was detected?

As suggested by the reviewer, we improved the figure (now Figure 2g) to show the percent of samples instead of the absolute number of samples in which a particular miRNA was detected.

- Figure 2i: It is difficult to understand what is shown.

We apologize that the upset plot in this figure (now Figure 2h) was not described sufficiently. We now improved the description in the results section as follows (lines 211-216):

Finally, a set analysis comparing the overlaps of all detected miRNAs per protocol was performed. This underlined the high heterogeneity of miRNAs detected per experimental set-up. While 69 miRNAs were detected in at least one replicate in all protocols, 60 miRNAs were exclusively detected by the SBN protocol, followed by the SB protocol with 57 exclusive miRNAs and the SBN_CL protocol with 53 exclusive miRNAs (Fig. 2h).

We also revised the legend for this plot (lines 733-739):

Upset plot showing the miRNAs jointly detected by multiple protocols, or exclusively found in only one protocol (orange). The bar plot at the top shows on the y-axis the number of miRNAs detected by the protocols highlighted by connected black or orange dots in the grid below. The bar plot on the left shows on the x-axis the total number of miRNAs detected in a least one of the replicates of the protocol shown on the y-axis.

- Methods section: During the density gradient centrifugation a 1200g was used, which is quite high. Is this correct?

The reviewer is right, the density gradient centrifugation was conducted at only 1000g. This typing mistake was corrected (line 608).

Reviewer #2 (Remarks to the Author):

In this paper, the authors benchmark protocols for the detection of micro RNAs in single cells. To this end, they adapt four different ligation based protocols with variations resulting in a total of 18 protocols plus one polyAdenylation based protocol.

In first testing round 1pg of the microRNA reference panel miRExplore was spiked into single cell equivalents from the cell line MCF7 cell-line and then sequenced in triplicates following each of the 19 protocols being evaluated. The main criteria, were the cDNA yield & size ranges from the bioanalyzer, the number of reads mapping to known miRs and the number of spike in miRs detected. Based on this the field is narrowed to 8 protocols that are now evaluated using real single cell data again in 3 replicates, i.e. only 3 cells. The final method comparison is done using patient data with at the most only 28 cells per patient.

Major things:

1. The manuscript could benefit from a better description of the miRNAs detected and in the case for which the reference panel is available, contrast them with the ones that were not detected. For Readers, like me, who are not familiar with miRExplore it would also be great to outline the basic properties: How many? Relative concentration ...

We appreciate this great idea. As we elaborate below and in the revised manuscript, the analysis suggested by the reviewer highlighted quite interesting results. As the reviewer suggests, miRXplore is well suited since we could indeed expect that all miRNAs that are included in the panel are detected. Following the suggestion of the reviewer we added Supplementary Table 5 that contains each sequence in the panel along with the information in which of the now 80 experiments it was detected. This table is sorted according to the number of experiments showing the miRNA in decreasing order. On top are those that are detected in all experiments, on bottom those that are not detected at all (new Supplementary Table 5; please see also our response to comment 4 of the same reviewer). Finally, we compared the nucleotide composition of those miRNAs that are in the first group and in the latter group and compare them to each other. We observed that the group of sequences that were always detected had substantially higher guanine content while those not detected had lower guanine content (Supplementary Figure 5). Moreover, we recognized that those sequences that were detected at low frequencies independent of the used protocol are indeed no true biological sequences but calibration oligonucleotides. We highlight these findings in the revised manuscript (lines 143-149), added the afore mentioned supplemental table with all details and added a figure showing the correlation of percentage guanine versus the frequency of detection for a sequence.

Following the suggestion of the reviewer we also provide a description of the miRXplore Universal reference (Miltenyi) in the new Supplementary Table 2.

2. The initial selection is based on the consistency of 3 replicates, whereas each replicate is a single cell equivalent. In my opinion this is too little, the entire evaluation here might hinge on a single small pipetting inconsistency. Furthermore, including more replicates would allow the authors to provide better statistics. For example they could evaluate the impact of all protocol modifications in a generalised linear model or GC content and secondary structure as the authors suggest in the Discussion.

We agree with the reviewer that the number of replicates is a minimum. Therefore, we doubled the number of replicates in all stages of the protocol evaluation process. Already in the original manuscript we included 79 miRNA libraries (56 in stage 1, and 23 in stage 2). We have now increased this number to 127 libraries (80 in stage 1 and 47 in stage 2). Of course, we repeated all computations and updated all figures. As you can see from the revised manuscript, the results did not change in their essence. But, as the reviewer suggests, the credibility and stability of the data has now improved substantially.

We followed the reviewer's suggestion and investigated for the miRXplore spike-in (stage 1) the influence of the sequence nucleotide content on the miRNA detection rate and confirmed the G-content as a potential source of variance (Supplementary Fig. 5). We also investigated the secondary structure indirectly by evaluating the minimum free energy of each sequence. Although several publications show that also miRNA secondary structure can influence the quantification accuracy, we did not find any evidence in our data (Supplementary Table 5).

3. The statistics often don't match the conclusions: How can infer bias from a coefficient of variation, bias refers to a consistent under or overestimation due to technical problems, not to an increase in variance. Similarly, spearman's correlation coefficient is not a good measure of reproducibility. The ordering of expression levels of genes and I guess also miRs is often very robust towards deviations.

We apologize the inadequate wording. With the coefficient of variation, we indeed aimed to show the stability of results rather than to infer anything on potential biases. In contrast, a coefficient of variation could show a bias towards increased variance. If for protocol A the coefficient of variance across all miRNAs is substantially lower than the coefficient of variation for a second protocol B, protocol B can be considered biased to higher variance. We corrected the manuscript in this regard.

As the reviewer points correctly, we should have used a different measure to assess the reproducibility. Therefore, we now base our conclusions on the Euclidean distance of the log₂ transformed expression values, thereby taking into account the expression level deviations, which were not covered by correlation coefficients (Figure 1e, Figure 2f).

4. Also due to the low sample sizes, the results remain rather descriptive. For example the authors report a number of miRs that are unique to one replicate and others that were found everywhere. However, the authors do not investigate this observation further. In my opinion it might well be that such a result is expected. This could be modelled by analysing the sampling distribution.

We apologize for the lacking clarity of Figure 2i (now Figure 2h), which was also not clear for reviewer 1. We did not focus on single replicates but considered all replicates of one protocol together to determine which miRNAs were unique to one protocol, since this approach is much more stable than focusing on the replicates separately, as you correctly pointed out. With the increased number of replicates per protocol the power of this analysis is further increased. In this regard we focused most on the miRXplore experiments, as suggested already in the first comment of the reviewer. We analyzed which miRNAs were detected by which protocol.

5. At the end a clear summary, maybe as a figure, that shows the different criteria and the performance of the tested methods is lacking.

We decided not to add an additional figure, but we changed several parts of the manuscript to mitigate the reviewer's critic. Figure 1a was improved to show more clearly the different experimental stages and the applied selection criteria. Respectively, the text of the results chapter experimental design was adapted and the selection criteria were pointed out (lines 87-95).

All in all, I think the sample sizes and the variable design of the different tests is not very informative for a method comparison paper.

We believe that the increase in sample size from 148 to 244 and the harmonization of the tests significantly improves the information content of the study and now also meets the requirements of a method comparison paper.

Reviewers' Comments:

Reviewer #1:

Remarks to the Author:

The authors have done an excellent job in revising the manuscript. All the points I had raised have been sufficiently addressed and I have no further comments.

Reviewer #2:

Remarks to the Author:

The authors have addressed all my concerns: the sample sizes were increased to a more appropriate level, a discussion of the miRExplore reference was added and the statistics were adjusted to fit the questions.